# Machine Learning Algorithms Provide Greater Prediction of Response to SCS Than Lead Screening Trial: A Predictive AI-Based Multicenter Study

**DOI:** 10.3390/jcm10204764

**Published:** 2021-10-18

**Authors:** Amine Ounajim, Maxime Billot, Lisa Goudman, Pierre-Yves Louis, Yousri Slaoui, Manuel Roulaud, Bénédicte Bouche, Philippe Page, Bertille Lorgeoux, Sandrine Baron, Nihel Adjali, Kevin Nivole, Nicolas Naiditch, Chantal Wood, Raphaël Rigoard, Romain David, Maarten Moens, Philippe Rigoard

**Affiliations:** 1PRISMATICS Lab (Predictive Research in Spine/Neuromodulation Management and Thoracic Innovation/Cardiac Surgery), Poitiers University Hospital, 86021 Poitiers, France; Maxime.BILLOT@chu-poitiers.fr (M.B.); Manuel.ROULAUD@chu-poitiers.fr (M.R.); dr.bouche@gmail.com (B.B.); Bertille.LORGEOUX@chu-poitiers.fr (B.L.); Sandrine.BARON@chu-poitiers.fr (S.B.); Nihel.ADJALI@chu-poitiers.fr (N.A.); Kevin.NIVOLE@chu-poitiers.fr (K.N.); nicolas.naiditch@gmail.com (N.N.); chantalwood@orange.fr (C.W.); romain-david@hotmail.fr (R.D.); Philippe.RIGOARD@chu-poitiers.fr (P.R.); 2Laboratoire de Mathématiques et Applications, UMR 7348, Poitiers University and CNRS, 86000 Poitiers, France; yousri.slaoui@math.univ-poitiers.fr; 3Department of Neurosurgery, Universitair Ziekenhuis Brussel, 1090 Brussels, Belgium; lisa.goudman@gmail.com (L.G.); mtmoens@gmail.com (M.M.); 4STUMULUS Research Group, Vrije Universiteit Brussel, 1090 Brussels, Belgium; 5AgroSup Dijon, PAM UMR 02.102, Université Bourgogne Franche-Comté, 21000 Dijon, France; pierre-yves.louis@agrosupdijon.fr; 6Institut de Mathématiques de Bourgogne, UMR 5584 CNRS, Université Bourgogne Franche-Comté, 21000 Dijon, France; 7Department of Spine Surgery & Neuromodulation, Poitiers University Hospital, 86021 Poitiers, France; Philippe.PAGE@chu-poitiers.fr; 8Dyname, UMR 7367, Faculty of Social Sciences, University of Strasbourg, 67083 Strasbourg, France; 9CEA Cadarache, Département de Support Technique et Gestion, Service des Technologies de L’Information et de la Communication, 13108 Saint-Paul-Lez-Durance, France; cydran@gmail.com; 10Physical and Rehabilitation Medicine Unit, Poitiers University Hospital, University of Poitiers, 86021 Poitiers, France; 11Prismatics Lab & Spine Surgery and Neuromodulation Department, Poitiers University Hospital, 86021 Poitiers, France

**Keywords:** spinal cord stimulation, screening trial, lead trial, infection, supervised learning, machine learning, predictive modeling, patient outcome

## Abstract

Persistent pain after spinal surgery can be successfully addressed by spinal cord stimulation (SCS). International guidelines strongly recommend that a lead trial be performed before any permanent implantation. Recent clinical data highlight some major limitations of this approach. First, it appears that patient outco mes, with or without lead trial, are similar. In contrast, during trialing, infection rate drops drastically within time and can compromise the therapy. Using composite pain assessment experience and previous research, we hypothesized that machine learning models could be robust screening tools and reliable predictors of long-term SCS efficacy. We developed several algorithms including logistic regression, regularized logistic regression (RLR), naive Bayes classifier, artificial neural networks, random forest and gradient-boosted trees to test this hypothesis and to perform internal and external validations, the objective being to confront model predictions with lead trial results using a 1-year composite outcome from 103 patients. While almost all models have demonstrated superiority on lead trialing, the RLR model appears to represent the best compromise between complexity and interpretability in the prediction of SCS efficacy. These results underscore the need to use AI-based predictive medicine, as a synergistic mathematical approach, aimed at helping implanters to optimize their clinical choices on daily practice.

## 1. Introduction

Failed back surgery syndrome (FBSS), now known as persistent spinal pain syndrome type 2 (PSPS-T2), is characterized by persisting back and/or leg pain despite one or several spinal surgical procedures [1,2,3,4]. PSPS-T2 incidence remains devastating, affecting 10 to 40% of operated spine patients [5,6]. This generates severe social [7], financial and psychological burdens for a significant number of patients [8]. Given this context, PSPS-T2 patients are referred to a large panel of therapies through multidisciplinary team pain management, and when refractory, they can be successfully treated with spinal cord stimulation (SCS) [9,10,11,12,13,14]. SCS outcomes rely on patient selection, which remains challenging, since implanters have to face the nature of pain, infiltrating all dimensions of patient quality of life [15], defining different trajectories for different patient profiles and impacting the most vulnerable refractory PSPS patients, potentially eligible to SCS, with an extreme variety of clinical presentations.

With a legitimate determination to bring some medico-economic rationale to technological evolution, healthcare systems provide strict rules of SCS implantation since its initial diffusion; following international recommendations, a lead trial must be performed and validated before any permanent device implantation [16,17,18,19]. In addition, SCS lead trialing is intended to potentially optimize neural structure spatial targeting, as some implanters choose to perform intra-operative table testing and/or deliberate temporary trialing to optimize lead placement and to reimplant the same patient later permanently, when test results are positive [20]. Moreover, from a patient’s perspective, lead trialing offers an opportunity to “mimic” the potential added value of SCS during an average 5–10-day period of the trial [21].

However, recent clinical data highlight some major limitations of this approach: -First, one can observe a lack of homogeneity among practices, making comparisons and recommendations regarding lead trialing very difficult: a patient can be trialed with a surgical lead (requiring an invasive approach) vs. a percutaneous trial, where one, two or three percutaneous lead(s) are placed in the spinal canal, under local sedation or general anesthesia [14,18]; the patient can be trialed from 3 to 28 days, depending on the healthcare system [22], which is responsible for significant differences in terms of lead trial complication rates [14]; during the trial, patients can test one, a few or hundreds of programs [18], which might affect the lead trial outcome, depending on the clinical expertise of the implanting team and trial duration. Ultimately, lead trialing could create a bias, since the trial cannot be blinded except in paresthesia-free techniques at the price of strict protocols, which cannot be standardized among practices, centers and countries, except for research purposes.-Lead trial outcomes have been reported using the Visual Analogue Scale (VAS) or Numeric Pain Rating Scale (NPRS). Based on these assessments, guidelines recommend that a refractory PSPS patient can be eligible to permanent implant, if a 30–50% pain decrease is observed during the trial [14,16,17,19]. It appears that this unidimensional modality of pain assessment can no longer be considered as the only gold standard to delineate the implanting physician’s guidelines and international recommendations, since this would reflect only one dimension of the patient’s quality of life, needs and expectations [23]. While composite multidimensional pain indexes, following the application of pain therapy on chronic refractory patients [23], would help to capture the essence of pain substrate and pain potential relief, they are not currently part of the recommended pain assessment toolbox.-Third, it appears that patient selection, final implantation ratio and patient outcomes, with or without lead trial, are similar [24]. Indeed, several studies have evaluated the ability of the screening trial to predict the long-term efficacy of SCS [24,25,26]. In a multicentric randomized controlled trial, Eldabe et al. [24] compared pain relief at 6-month follow-up between 54 patients who underwent a screening trial and 51 patients who did not. They found no significant difference regarding pain relief between groups. Furthermore, they reported that the diagnostic accuracy of the screening trial presented sensitivity of 100% (percentage of patients with a positive screening trial among responders) and specificity of 17% (percentage of patients with a negative screening trial among non-responders). All in all, the authors concluded that outcomes do not differ with or without the screening trial, and that the screening trial was not able to identify long-term non-responders.-Fourth, it has been shown that infection rates during SCS trials increase logarithmically within time, after 14 days of trialing, and can compromise the therapy [14].

To date, we can conclude there is no consensus on the usefulness of the screening trial phase due to the invasiveness of the procedure and potential complications such as lead migration, dural puncture during lead placement and/or infection [27,28]. Aiming to find the best compromise between risks and benefits, several authors have attempted to find alternatives to screening trial, studying factors such as age [29,30], sex [31,32], psychological state [30,33,34,35] and pain duration [36,37], their objective being to demonstrate that they are associated with long-term pain relief following SCS. However, only classical statistical methods including logistic regression models and correlation analyses have been applied to identify these predictors, which renders their transposability to daily practice quite difficult, if not impossible. Recently, Goudman et al. showed that machine learning models can achieve good performance for predicting the efficacy of high-frequency SCS in patients with PSPS-T2 [38], which is highly specific and only applicable to high-frequency SCS; this represented a first step at the use of predictive medicine in this field.

Using our experience of multidimensional composite pain assessment and previous predictive research [15,23], we hypothesized that machine learning models, based on homogeneous data pooled from several multicenter studies, could be used as robust screening tools to predict SCS efficacy. To test this hypothesis, we developed, for the purpose of this study, several algorithms designed to confront model predictions and to compare our model performances to lead trial results among 103 implanted patients.

## 2. Materials and Methods

### 2.1. Patient Data

Data from two different prospective comparative studies were used to conduct this work. 

#### 2.1.1. First Dataset

The first study is ESTIMET [18], which is a multicenter randomized controlled trial, including 115 PSPS-T2 patients eligible for SCS and implanted with surgical multicolumn SCS paddle lead, in 12 French centers with a 1-year follow-up. The study details are available at https://clinicaltrials.gov/ct2/show/NCT01628237 (accessed on 12 March 2021). The primary objective of this study was to compare the efficacy of multicolumn SCS programming to the efficacy of monocolumn SCS programming. As part of the ESTIMET study, all subjects provided informed consent and enrolled in the following ethical committee approval (CPP-Ouest III) [18]. The study population consisted of PSPS-T2 patients with refractory pain, eligible to SCS according to the French guidelines for SCS selection and implantation. Per these guidelines, an average of 7-day screening trial period was mandatory for all study patients. Patients with a 50% pain decrease, or patients for whom the improvement was clinically important according to a patient–implanter agreement, were implanted with a permanent SCS device at the end of the trial. Among the ESTIMET study patients, those who did not try transcutaneous electrical nerve stimulation (TENS) were removed from the analysis because TENS efficacy belongs to the set of predictive variables used in the development of the models in this paper (Figure 1). Finally, ninety-one patients who underwent TENS therapy, completed baseline data and completed the study 1-year follow-up were included in the analysis. ESTIMET study data were used for the training and internal validation process of our models.

#### 2.1.2. Second Dataset

The second study is AIVOC (https://clinicaltrials.gov/ct2/show/NCT02821897 (accessed on 12 March 2021)), which is a monocentric comparative study, including 15 patients implanted with SCS under general vs. awake anesthesia at Poitiers University Hospital (France), with a 1-year follow-up. This study examines the effect of target-controlled intravenous infusion on SCS implantation, lead placement optimization using patient intra-operative feedback and SCS efficacy on back pain coverage. Patients in this study were randomized to either be implanted using general anesthesia during lead implantation or to be implanted using target-controlled intravenous anesthesia with active patient–implanter cooperation during the surgery. Three patients were lost to follow-up. The 12 remaining patients who underwent 12-month follow-up from the AIVOC study were used for external validation. 

For both the ESTIMET and the AIVOC studies, following the verification of inclusion/non-inclusion criteria, patients were included and evaluated at baseline, in terms of their sociodemographic, psychological, radiological and clinical characteristics. One month after inclusion, all patients were implanted with SCS and underwent a permanent trial: an average of 7-day screening trial period was mandatory for all, per French recommendations of the Ministry of Health. Patients with a 50% pain decrease, or patients for whom the improvement was clinically important according to a patient–implanter agreement, were implanted with a permanent SCS device at the end of the trial [19].

### 2.2. Studied Variables

#### 2.2.1. Primary Outcome

We evaluated SCS efficacy using pain intensity assessed by means of a Visual Analogic Scale (VAS), health-related quality of life (EuroQol with five dimensions and three levels (EQ5D-3L)) [39], functional disability evaluated using the Oswestry Disability Index (ODI) [40] and severity of depression using the Montgomery and Asberg Depression Rating Scale (MADRS) [41]. In order to achieve holistic evaluation, we used principal component analysis (PCA) with one principal component, including the percentage of global VAS decrease (i.e., (VAS baseline-VAS 12-month)/VAS at baseline), percentage of EQ-5D increase, percentage of ODI decrease and percentage of MADRS decrease between baseline and 12-month follow-up. The first principal component of the PCA was taken as a standardized Global Health Improvement Score (GHIS). Patients were considered as responders if they had a GHIS ≥ 0. Patients who had a negative screening trial or a GHIS < 0 were considered as non-responders. This outcome was used as a binary dependent variable in our SCS efficacy classification problem. We also evaluated the relationship between this composite outcome and the standard outcomes used in SCS literature and recommended in France by health authorities, which are 50% pain VAS decrease, 30% ODI decrease and 0.19-point EQ-5D index difference.

#### 2.2.2. Predictors

To avoid any bias induced by variable selection based on statistical significance, no primary variable selection was conducted. We used the 14 variables that were the most widely studied in the SCS literature. They included age [29,30], sex [31], depression score [30,34] measured using MADRS score, body mass index (BMI) [42], pain syndromes associated with nervous or somatic lesions (hypoesthesia, brush allodynia) [43], pain increase by movement or by sustained position, TENS efficacy [44], baseline EQ-5D index, baseline back and leg VAS, baseline ODI score, pain duration (in years) [36,37] and the Medication Quantification Scale (MQS III) to measure medication consumption in chronic pain [45].

### 2.3. Statistical Methods

The statistical analyses were performed using R 3.6.0 software (R Foundation for Statistical Computing, Vienna, Austria).

#### 2.3.1. Descriptive Analysis

Categorical variables were described by numbers and percentages, while quantitative variables were described by their means and standard deviations (SD) or by their median and interquartile range (IQR) depending on the skewness of the variable. No missing data imputation was performed.

#### 2.3.2. Multivariate Analysis

For this analysis, we developed several algorithms based on logistic regression (LR), regularized logistic regression (RLR), naive Bayes (NB) classifier, artificial neural networks (ANN), support vector machines (SVM), classification and regression trees (CART), random forest (RF) and gradient-boosted trees (GBT) to test our hypothesis. Commonly used model performance metrics ROC curve and area under ROC curve (AUC) were used to evaluate model accuracy. ESTIMET study data (*n* = 91) were used as a training set, and AIVOC study data (*n* = 12) were used as an independent testing set to confront model predictions and to compare model performances to the 5- to 10-day trial results. Leave-one-out cross-validation and Monte Carlo cross-validation were used to assess internal validity and performance variability.

All data were standardized in order to facilitate interpretation and convergence of the models. Standardization was conducted by subtracting the mean and dividing by the standard deviation.

In this section, each model and its implementation are described briefly. The following classification models were used in this paper to predict SCS outcome at 12-month follow-up:

LR: This model was developed using the *glm* function available in the R stats package. Even if logistic regression can only detect linear relations between variables, it is still widely used because of its simplicity and interpretability, and it has shown better performance on simple classification problems where classes can be linearly separated. The backward–forward (bidirectional) stepwise variable selection procedure was used in order to identify the best LR model based on the Akaike information criterion (AIC). To avoid overfitting, no interaction terms were included in the model.

RLR: This regularized generalized linear model [46] was developed using the *glmnet* package. Regularization is a technique used to shrink or reduce insignificant effects in the logistic regression to zero. This technique enables the model to avoid overfitting because it reduces model variance. As our data contain few variables, we opted for the use of ridge regularization.

SVM: Support vector machine models are known for their classification capability, since SVM algorithms are computationally stable and generalize well, giving a sufficient number of training examples [47]. Besides linear relations, SVM can detect nonlinearities by transforming input data into a space of higher dimensionality using a kernel function. For the purposes of this study, we used the radial basis function kernel to allow nonlinearity. SVM was developed using the *svm* function available in the *e1071* R package [48]. The optimal cost and gamma parameters were identified through cross-validation.

NB: This classifier [49] was developed with default hyperparameters using the naive Bayes function available in the *e1071* R package.

ANN: The neural networks model [50] was developed using the *keras* package [51], which is a popular deep learning Python package that has been added recently to R software as a package available in the Comprehensive R Archive Network (CRAN). To avoid overfitting, we used a relatively small ANN. The ANN model contained two hidden layers, each comprising eight nodes. To ensure model convergence, we trained the ANN model on standardized data (0 mean and unit variance). Sigmoid activation functions were used to allow nonlinearity. The weights were estimated using the backpropagation method, which is a gradient-based optimization method. This allows error estimation at the output of the hidden layer neurons, thereby enabling the update of weights in the hidden layers by means of error gradients.

CART: The classification tree model was developed using the *rpart* function available in the package with the same name [52]. In a CART model, all patients initially belong to a simple node representing good and non-responder rates. Afterwards, the node is split, creating two new child nodes. The splitting is performed by choosing the predictor and the optimal split point (e.g., age > 45) that differentiates good responders from non-responders. The algorithm stops when the observations inside the nodes are homogeneous in terms of the outcome variable and further splits are undesirable. The minimum number of observations in a node was set at five.

RF: The random forest [53] was trained using the *randomForest* function available in the R package with the same name [54].

GBT: This gradient-boosting-based model [55] was developed using the *xgboost* package [56].

Both RF and GBT are tree-based ensemble models, which are a type of models based on the principle that averaging the predictions of several small models could help to obtain a better model. Each of the two models uses a different ensemble learning technique. RF uses bagging (bootstrap aggregating), which can be described as follows: for each iteration, a decision tree model is created using data from a bootstrap sample drawn from the training set, independently of other iterations. After growing all the trees, each tree casts a unit vote for the outcome (good or non-responder) of a new observation. The final prediction for this observation is the average of predictions obtained from all the trees.

GBT, as the name suggests, uses a technique called boosting, whereby weighted combinations of decision trees are constructed into a stronger classifier in an iterative way (contrary to random forest, where the weights are uniform, and the trees are grown independently). The strongest classification tree is weighted to count more substantially in the prediction of outcome. A tree that most accurately classifies examples that were misclassified by the first tree is grown next. This procedure allows the trees that are weak on some examples to be compensated by another tree, which performs better on the same examples.

### 2.4. Testing Data and Model Assessment

Cross-validation was used to identify optimal hyperparameters of the models and to assess their internal validity. Our cross-validation procedure goes as follows: We divide our training dataset into 10 separate datasets (10-fold cross-validation). One subset is kept for model assessment. Nine-fold cross-validation is conducted on the remaining nine subsets to identify the optimal hyperparameters, which are then used to develop the models. The final models are then tested on the 10th subset. This process is conducted for all of the 10 folds. This procedure allows us to reduce evaluation bias associated with identifying optimal hyperparameters and evaluating the models on the same subset.

The models and screening trial efficacy (percentage of pain decrease) were evaluated using sensitivity, specificity, accuracy, and area under ROC curve (AUC). The cross-validation means of these evaluation measures and their standard deviations are reported.

### 2.5. External Validation

An independent dataset of the 12 remaining patients from the AIVOC study was used for model assessment and external validation. Similarly, models were evaluated externally using sensitivity, specificity, accuracy, and area under ROC curve (AUC). 

## 3. Results

### 3.1. Descriptive Analysis

Descriptive statistics of our training and testing data can be found in Table 1. The majority of our predictors were homogeneous between the training and testing datasets. Of the training dataset, 49.5% were male and 50.5% were female, while 41.7% of the testing dataset were male and 58.3% were female. The mean age of our training sample was 47.7 (SD = 9.5) and the mean age of the testing sample was 49.5 (SD = 14.7). Of the patients, 42.8% had white-collar jobs and 15.4% were without professional activity. The majority of patients (65.9%) had stopped working due to chronic pain. The majority of patients had only one or two spinal surgeries (48.4% had one surgery and 26.4% had two surgeries). We observed some differences between the training and testing datasets. Patients in the training dataset were less likely to respond to TENS therapy than patients in the testing dataset (52.7% for training dataset vs. 83.3% for testing dataset). Pain medication consumption was also higher in the training dataset than the testing dataset (MQS of 24.5 (SD = 14.7) vs. 5.6 (SD = 7.8)).

The results of this PCA leading to the development of our composite outcome can be found in Table 2. The first component of our PCA explained 50.1% of the total variance. The ODI percentage of decrease had the highest weight in the first component (0.86) followed by VAS (0.81), MADRS (0.59) and EQ-5D (0.51). In our training dataset, 45 patients (49.5%) had a positive holistic response to SCS and 46 (50.5%) had a negative holistic outcome. Similarly, six patients (50%) had a positive outcome and six patients (50%) had a negative outcome in the testing dataset. Table 3 shows the relationship between our composite outcome and the classical pain assessment outcomes used in the literature.

### 3.2. Training Data Results (Internal Validation)

We developed our models using the eight different binary classification methods described in Section 2.3. The SVM model showed the highest performance metrics according to our cross-validation procedure results (AUC = 0.801; SD = 0.202). It had a specificity of 81.3% (SD = 14.8%) and sensitivity of 80.7% (SD = 20.1%). The GBT model also showed good performances with lower variability between folds, which indicates a more stable model (AUC = 0.790 (SD = 0.105); specificity = 80.0% (SD = 3.6%); sensitivity = 70.1% (SD = 16.7%)). The two logistic regression models, LR and RLR, showed results comparable to the previous models. The regularized logistic regression model had an AUC of 0.781 (SD = 0.120), sensitivity of 69.8% (SD = 12.0%) and specificity of 73.0% (SD = 12.9%). Our logistic regression model had an AUC of 0.779 (SD = 0.114), sensitivity of 70.9% (SD = 18.0%) and specificity of 72.8% (SD = 13.3%). The AUCs of the RF, naive Bayes and CART were 0.755 (SD = 0.123), 0.697 (SD = 0.153) and 0.657 (SD = 0.136), respectively. According to these results, we would recommend GBT, SVM or RLR models. The advantage of the RLR model is that it can be interpreted as a simple logistic regression model. The AUC of the screening trial was 0.670 with a sensitivity of 79.5% and specificity of 52.4%. 

### 3.3. External Validation

The results of our models using the external validation set can be found in Table 4. The results obtained on the external validation set were similar to those on the external set. The best performances were achieved using the RF model, the GBT model and the RLR model. While the RF model showed lower performance on the training set, the GBT and RLR model had good results on both the training and testing sets. Similarly to the training set, the testing set screening trial results showed good sensitivity (100%) but bad specificity (33.3%). Forty percent of patients with a bad 1-year outcome had a 50% pain decrease following the screening trial. 

We also evaluated patients for whom the model predictions deviate heavily from their true outcome. Among patients with a prediction smaller than 20% (the model is very confident in their negative response to SCS), none had a positive outcome, which coincides with their predictions. On the other hand, among patients with a prediction greater than 80% (the model is very confident in their positive response), two patients had a negative outcome. The first patient is a female (aged 70 years) with back and leg pain. Following spinal cord stimulation, the patient observed a significant decrease in leg pain and a moderate decrease in back pain. At 12-month follow-up, the leg pain relief was still significant but her back pain was not (severe pain). The patient kept focusing on her back pain, which altered her patient-reported outcomes. The patient was very satisfied with her stimulation device efficacy, in general. We understood that the residual back pain component was mechanical and implanted her with a subcutaneous lead after the end of the study, as a salvage therapy. She achieved significant back pain relief. The second patient was a 39-year-old female who was implanted with spinal cord stimulation 2 years after pain onset to treat back and leg pain. Similarly to the first case, this patient observed moderate leg pain relief without sufficient back pain relief. The patient was implanted with a subcutaneous lead, as a hybrid therapy, to address her back pain component, after the end of study. She started professional retraining for a new job.

### 3.4. Model Interpretability

The majority of the models discussed and analyzed in this paper are black box models, meaning that it is difficult to extract useful information on how the variables interact with SCS outcomes. However, this interpretability is sometimes disregarded in order to achieve more complexity. In this section, we will show the role of explanatory variables in decision making for the logistic regression model. Table 5 shows the unstandardized coefficients, standardized coefficients and their 95% confidence intervals, for each variable.

Based on the logistic regression model results (Table 5), the baseline depression score had a significant effect on the outcome of the treatment (MADRS odds ratio = 0.908; *p*-value = 0.002). Depressive patients have a reduced chance of having a good outcome 1 year after the implantation of SCS device. Interestingly, we observed a greater probability of achieving a successful outcome not only in patients with hypoesthesia related to back pain (odds ratio = 10.59; *p* = 0.0008) but also in those with positional back pain symptoms (odds ratio = 4.48; *p* = 0.043). Patients who achieved a 50% pain decrease after TENS therapy before SCS had a better chance of successful SCS therapy at 1-year follow-up (odds ratio = 0.2691544; *p* value = 0.016).

## 4. Discussion

The goal of this paper was to investigate the comparative performance of machine learning models vs. recommended lead screening trials to predict SCS efficacy. Our results show that machine learning techniques offer the opportunity to predict patient SCS long-term outcomes with higher accuracy than SCS trialing (when considering a failed SCS trial as a negative long-term response). Compared to previous research works aiming to identify SCS outcome predictors, the originality of this mathematical approach is based on computing these predictors in a multivariate statistical model in order to transpose this predictive model to PSPS-T2 patient real-life conditions [45]. This study also led us to test our multi-dimensional pain assessment approach, reflected in a composite index, based on principal component analysis conducted on pain intensity, functional disability, quality of life and psychological distress measures. We have shown that this holistic pain evaluation is significantly associated with these four pain dimensions.

### 4.1. Potential Added Value of a Predictive Model vs. Lead Trialing

Applying invasive therapeutic strategies to vulnerable patients implies empiric choices for the implanting physician. To develop a strong therapeutic alliance, when convinced, the physician also has to convince his patient that the benefits/risk ratio is in favor of applying a new technique. In that sense, a lead screening trial appears very comfortable, not only for the patient but also for the physician, as described with humor by an internationally renowned implanter: “The beauty of SCS is that you try it before you buy it…!”.

Conversely, it is now well documented that lead trialing is associated with substantial complications [24,26], including infection [57], which can turn from what one patient has described as a “honeymoon” into a nightmare, leading to SCS explantation (11 out of 108 patients implanted with a multicolumn lead in ESTIMET study [18]).

Reconsidering back the benefits/risks balance, one should examine the benefits of lead trialing carefully, if risks are considered as substantial. Eldabe et al.’s recent study leads to implanter disappointment, by showing that SCS outcomes remain similar, whether using lead trialing to select patients for permanent implantation or not [24]. Moreover, in a recent RCT, Thomson et al. demonstrated that patient preference would be in favor of a single-step approach. 

Therefore, an original mathematical predictive approach could aspire to replace SCS trial procedure with a more objective, noninvasive and accurate strategy based on AI, using machine learning models to reduce the risks of complications associated with invasive surgical procedures by more efficiently selecting the patients who may benefit from spinal cord stimulation. 

### 4.2. Machine Learning Model Accuracy to Predict SCS Efficacy

We more accurately predicted the efficacy of SCS than the screening trial (AUC = 0.69 for screening trial vs. AUC = 0.81 for the RLR model). As a starting point, easy to transpose in daily practice, we recommend the use of RLR model, as it achieved a remarkable performance for both internal and external validation. In contrast with NB, RF and GBT showing even greater performance, the RLR model is also easily interpretable (it can be interpreted similarly to a logistic regression model), and variable selection is included in the model estimation procedure, which can simplify the creation of an automated decision tool. 

Few authors have proposed models predicting SCS efficacy. The most recent and relevant paper was published by Goudman et al. [45]. In their paper, they developed a logistic regression model for predicting high-dose SCS efficacy using data from 92 FBSS patients and a set of variables including age, sex, back and leg pain intensity, MQS, ODI, Pittsburgh sleep quality index, EQ-5D and second-order interactions between these variables. They achieved 90% specificity and sensitivity on an out-of-sample dataset consisting of 20% of their dataset. The model contained some very large coefficients compared to the scale of the variables. This might be due to the inclusion of a large number of interaction terms for an intermediate population sample size. Cross-validation results and screening trial predictive value were not reported in this paper despite very good performances to detect high-frequency SCS responders. Sparkes et al. [35] also proposed a logistic regression model for predicting a pain decrease at 12 months of follow-up using data from 56 FBSS patients. Their model included age, sex, duration of pain, anxiety and depression scores and coping strategies. These results need, however, to be taken with caution, due to the lack of out-of-sample validation.

Using a different strategy, Thomson et al. [21] proposed an SCS decision tool, based on consensus recommendations from a panel of experts. This concept should be considered as a complementary approach, reflecting key opinion leaders’ views, from a physician’s perspective, as discussed above, and not as an opposed vision. One approach is based on perception and convictions; “artistic science”, which characterizes the patient–physician relationship, is designed to facilitate the best therapeutic decision in a non-empiric world. The other approach is based on mathematical accuracy; “art of science” should help physicians to reinforce their choices and convictions, based on robust and quantitative substrate. While SCS literature on outcome predictors is currently heterogeneous, data/evidence-based medicine is clearly needed to establish new insights in the SCS community.

### 4.3. SCS Predictors of Lead Trial Success and SCS Long-Term Outcomes

Our logistic regression model shows that patients with higher levels of depressive symptoms are less likely to benefit from SCS, whereas patients with a higher perceived health-related quality of life (EQ-5D) are more likely to achieve a good outcome following SCS. This corroborates our findings from a previous work identifying two distinct patient profiles [15], for which depression was associated with a lower level of activity, resulting in lower quality of life. These profiles are considered as pejorative SCS outcome predictors in a recent sociological paper focusing on socio-professional status of PSPS patients [58].

TENS efficacy was also significantly associated with SCS efficacy. This has been shown since 2011 in a prospective predictive study published by Mathew et al. 

Lastly, back pain hypoesthesia and changes in pain depending on patient position are associated with a greater likelihood of a good outcome following SCS. These findings are complex to extrapolate. Back pain hypoesthesia is one main criterion of the DN4 (Douleur Neuropathique en 4 questions) questionnaire [59], confirming that SCS is an appropriate tool to address neuropathic pain [10]. The positional exacerbation of pain might indicate that there is a mix of neuropathic and mechanical pain components, which are associated in this PSPS patient population, as opposed to continuous pain, preventing patients from participation in daily activities. This could be considered as an indirect marker of patient willingness to maintain a certain level of activity, and as detailed in our paper focusing on socio-professional status and patient activity [58], we found that SCS outcomes improved when patients were able to develop adaptive coping strategies with pain, to avoid kinesiophobia and seek functional capacity preservation. These results finally suggest that psychological evaluation and pain typology are important in patient selection prior to SCS implantation.

### 4.4. Study Strengths and Limitations

Apart from the originality of combining several machine learning models to reach our objectives, this study has several strengths, such as: -A double dataset, extracted from prospective comparative studies, comprising a large sample size, to reduce biases typically associated with this type of research and to ensure the maximal potential of generalization of our predictive models.-The multicenter nature of our sample also helps to ensure generalizability and applicability to clinical practice.-The development of a composite outcome, based on objective methods, which might lead to optimize patient satisfaction evaluation and, therefore, more precisely capture patient needs and expectations so as to define what should be considered as a positive outcome. However, two patients presented positive predictions related to significant “negative” results. While these patients had good outcomes for leg pain, back pain still remained unrelieved, altering the results of the global pain assessment. This highlights the relevance of assessing both global pain and individual pain areas in order to avoid misinterpretation of the overall assessment [60].

We also observed substantial limitations, which could be addressed by further research.

#### 4.4.1. Paresthesia Intolerance

One of the reasons patients might fail a conventional tonic SCS trial is their inability to tolerate SCS-induced paresthesia. Patient potential intolerance to SCS-induced paresthesia was not taken into consideration in these study outcomes. In contrast, it has been shown that there is a high correlation between intolerance to tonic SCS and intolerance to TENS-induced paresthesia [44]. Therefore, combining a TENS trial with our machine learning model could serve as a noninvasive multiplexed screening tool prior to SCS permanent implantation for conventional SCS. However, when applying new paresthesia-free neurostimulation waveforms, such as high-frequency, high-density and BURST stimulation, this notion would no longer remain relevant to analyze.

#### 4.4.2. Country-Dependent Variety of SCS Practices

Even though our data were extracted from several studies, including one multicenter national study with 12 centers, a main limitation would concern the relative homogeneity of practices across countries, such as lead choice, permanent trialing available in France and Europe, lead positioning and lead programming, depending on local cultures, which might not offer global extrapolations. An interesting way to overcome this limitation would be to pool and analyze data from large international registries, applying our machine learning model strategy, in further collaborative works. 

#### 4.4.3. Large-Scale Validation

Although we have shown that our model’s predictive power is superior to the screening trial in this sample of 103 SCS implanted PSPS patients, it will remain necessary to conduct a more robust comparative study to assess the overall superiority of utilizing machine learning models as opposed to screening trial on a larger cohort of patients. The evaluation criteria of such a study should include not only the comparative evaluation of lead screening trial vs. mathematical models, but also medico-economical evaluations, extrapolated from models through healthcare-based medicine and prospectively reported SCS implantation-related adverse events. 

## 5. Conclusions

Machine learning and statistical models appear to show potential interest to put traditional SCS lead trial clinical outcomes into perspective with AI-based predictions of SCS efficacy. The regularized logistic regression, random forest and gradient-boosted trees models have demonstrated the best performance and provided a good model fit to the testing data and a relatively good performance on training data. Almost all the proposed models show a better prediction power than lead trial outcomes for this specific population of PSPS patients. On a general aspect, these results reinforce the need for AI-based predictive medicine, with no other ambition than helping physicians to optimize their clinical choices, based on a synergistic mathematical approach. SCS applications are at an early stage and, by essence, remain limited. This deserves further study, based on high-fidelity composite multi-dimensional pain assessment and data extracted from large cohorts of implanted patients, to take this debate to another level. More specifically, in echo with recent clinical data, this study can be depicted as a seminal mathematical substrate to reconsider SCS lead trial utility. 

## Figures and Tables

**Figure 1 jcm-10-04764-f001:**
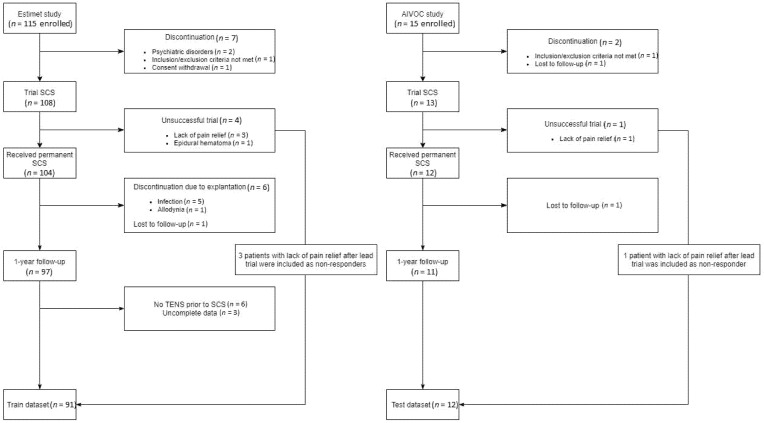
Flowchart of the patients who participated in ESTIMET and AIVOC included in this study.

**Table 1 jcm-10-04764-t001:** Descriptive statistics of our 12-month outcome and baseline characteristics for both the training and testing datasets.

Demographics	Train Set Descriptive Statistics	Test Set Descriptive Statistics
Profession		
Blue-collar job	9 (9.9%)	1 (8.3%)
White-collar job	39 (42.8%)	4 (33.3%)
Craftsman, shopkeeper, business	5 (5.5%)	2 (16.7%)
owner		
Executive	2 (2.2%)	0
Disability	6 (6.6%)	0
Intermediate profession	4 (4.4%)	0
Retired	6 (6.6%)	4 (33.3%)
Without professional activity	14 (15.4%)	1 (8.3%)
Other		
11	6 (6.6%)	0 (0%)
Work stopage		
Yes	60 (65.9%)	Not available
No	16 (17.6%)	Not available
Unemployed	8 (8.8%)	Not available
Retired and other	7 (7.7%)	Not available
**Pain history and management**		
Number of spinal surgeries		
1	44 (48.4%)	8 (66.7%)
2	24 (26.4%)	2 (16.7%)
3	13 (14.3%)	2 (16.7%)
4	6 (6.6%)	0 (0%)
5	3 (3.3%)	0 (0%)
Unknown	1 (1.1%)	0 (0%)
Type of spinal surgeries		
Decompression	69 (75.8%)	8 (66.7%)
Fusion	8 (8.8%)	2 (16.7%)
Decompression AND fusion	14 (15.4%)	0 (0%)
Unknown	0 (0%)	2 (16.7%)
Pain management with kinesiotherapy		
Yes	57 (62.6%)	Not available
No	34 (37.4%)	Not available
Pain management with osteopathy		
Yes	15 (16.5%)	Not available
No	76 (83.5%)	Not available
Pain management at a center of functional	rehabilitation	
Yes	11 (12.1%)	Not available
No	80 (87.9%)	Not available
Pain management using infiltrations		
Yes	35 (38.5%)	Not available
No	56 (61.5%)	Not available
**Response variable**		
Good composite outcome	45 (49.5%)	6 (50.0%)
Bad composite outcome	46 (50.5%)	6 (50.0%)
**Predictors at baseline**		
Age	47.7 (9.5)	49.5 (14.7)
Sex		
Male	45 (49.5%)	5 (41.7%)
Female	46 (50.5%)	7 (58.3%)
BMI	27.4 (5.04)	24.6 (3.9)
Pain duration	12.2 (10.7)	15.8 (15.1)
ODI	50.5 (9.1)	44.7 (12.8)
MADRS	16.9 (10.4)	11.3 (8.3)
EQ-5D	0.38 (0.20)	0.54 (0.19)
EQ-5D VAS	45.8 (17.3)	51.1 (20.0)
Leg VAS	75.0 (11.3)	72.9 (16.0)
Back VAS	71.2 (15.1)	67.6 (21.8)
TENS efficacy		
Effective	48 (52.7%)	10 (83.3%)
Not effective	43 (47.3%)	2 (16.7%)
Hypoesthesia		
Yes	28 (30.8%)	2 (16.7%)
No	63 (69.2%)	10 (83.3%)
Allodynia		
Yes	22 (24.2%)	5 (41.7%)
No	69 (75.8%)	7 (58.3%)
Positional pain changes		
Yes	74 (81.3%)	9 (75.0%)
No	17 (18.7%)	3 (25%)
MQS	24.5 (14.7)	5.6 (7.8)

BMI: body mass index, ODI: Oswestry Disability Index, MADRS: Montgomery–Asberg Depression Rating Scale, EQ-5D: EuroQol-5 Dimensions, VAS: Visual Analogic Scale, TENS: transcutaneous electrical nerve stimulation, MQS: Medication Quantification Scale.

**Table 2 jcm-10-04764-t002:** Composition of the first principal component of the PCA of our outcomes.

Variables Changes (%) between Baseline and 12 Months	1st Principal Component Loadings (50.1% of the Total Variance)
ODI	0.86
VAS	0.81
MADRS	0.59
EQ-5D score	0.51

ODI: Oswestry Disability Index, MADRS: Montgomery–Asberg Depression Rating Scale, EQ-5D: EuroQol-5 Dimensions, VAS: Visual Analogic Scale.

**Table 3 jcm-10-04764-t003:** Relationship between GHIS outcome and VAS decrease, ODI decrease and improvement in EQ-5D.

Outcomes	Good Composite Outcome (GHIS ≥ 0)	Bad Composite Outcome GHIS < 0
50% global VAS decrease		
Yes	43 (93.5%)	8
No	3	37 (82.2%)
30% ODI decrease		
Yes	34 (73.9%)	8
No	12	37 (82.2%)
0.19 points change in EQ-5D		
Yes	30 (65.2%)	17
No	16	28 (62.2%)

EQ-5D: EuroQol-5 Dimensions; ODI: Oswestry Disability Index; GHIS: Global Health Improvement Score; VAS: Visual Analogue Scale.

**Table 4 jcm-10-04764-t004:** AUC, specificity and sensitivity of screening trial and our model on the external validation set.

Model	True Good Outcome	True Bad Outcome
Screening trial (AUC = 0.69)		
Good outcome	6 (sensitivity = 100%)	4
Bad outcome	0	2 (specificity = 33.3%)
LR (AUC = 0.72)		
Predicted good outcome	5 (sensitivity = 83.3%)	2
Predicted bad outcome	1	4 (specificity = 66.7%)
RLR (AUC = 0.81)		
Predicted good outcome	5 (sensitivity = 83.3%)	2
Predicted bad outcome	1	4 (specificity = 66.7%)
SVM (AUC = 0.75)		
Predicted good outcome	6 (sensitivity = 100%)	2
Predicted bad outcome	0	4 (specificity = 66.7%)
NB (AUC = 0.81)		
Predicted good outcome	5 (sensitivity = 83.3%)	1
Predicted bad outcome	1	5 (specificity = 83.3%)
ANN (AUC = 0.72)		
Predicted good outcome	5 (sensitivity = 83.3%)	2
Predicted bad outcome	1	4 (specificity = 66.7%)
CART (AUC = 0.72)		
Predicted good outcome	4 (sensitivity = 66.7%)	1
Predicted bad outcome	2	5 (specificity = 83.3%)
RF (AUC = 0.83)		
Predicted good outcome	5 (sensitivity = 83.3%)	1
Predicted bad outcome	1	5 (specificity = 83.3%)
GBT (AUC = 0.81)		
Predicted good outcome	5 (sensitivity = 83.3%)	1
Predicted bad outcome	1	5 (specificity = 83.3%)

**Table 5 jcm-10-04764-t005:** Standardized coefficients of selected variables, confidence intervals and significance levels.

Variables	Unstandardized Coefficients (β)	Standardized Coefficients	95% CI	*p*-Value
Intercept	−3.044	−0.070	[−0.586; 0.447]	0.792
Duration of pain	−0.038	−0.041	[−0.939; 0.116]	0.137
MADRS	−0.097	−1.012	[−1.653; −0.371]	0.002 **
EQ5D VAS	0.032	0.554	[0.015; 1.093]	0.044 *
Leg VAS	0.040	0.449	[−0.09; 0.988]	0.102
Hypoesthesia: yes	2.361	1.096	[0.455; 1.737]	0.0008 ***
TENS: not effective	−1.312	−0.659	[−1.196; −0.122]	0.016 *
MQS	−0.024	−0.449	[−1.061; 0.163]	0.151
Positional pain changes: yes	1.500	0.588	[0.017; 1.159]	0.043 *

MADRS: Montgomery–Asberg Depression Rating Scale, EQ-5D: EuroQol-5 Dimensions, VAS: Visual Analogic Scale, TENS: transcutaneous electrical nerve stimulation, MQS: Medication Quantification Scale. * *p*-Value < 0.05, ** *p*-Value < 0.01, *** *p*-Value < 0.001.

## Data Availability

Not applicable.

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
