# Peer review of "Machine Learning Algorithms Provide Greater Prediction of Response to SCS Than Lead Screening Trial: A Predictive AI-Based Multicenter Study"

_jcm, 2021, doi:10.3390/jcm10204764_

Round 1

Reviewer 1 Report

I think this is a fascinating study, but I have a few questions.
Are the doctors and facilities seeing these patients the same? I would think that the compatibility between patient and doctor would be a significant factor in the treatment.

Were there any cases that deviated significantly from the predictive settings by the AI? I hope such patients do not exist, but I would like to know the elements of the patients whose results were off.

Reviewer 2 Report

Dear Authors

The title of this study seems to be consistent with the Journal of Clinical Medicine. I think this paper will be better if some minor and major points are corrected.

Minor points

The 'n' for the number of people or something should be in italics.

Except for %, all symbols, whether numbers or words, must be spaced by a space.

Major points

The reason for the continued appearance of FBSS may be the effect of surgical treatment, but it will also include various demographic characteristics of the patient and problems with postoperative management. Therefore, it is thought that this manuscript will be clearer if the demographic characteristics of the patients before or after surgery and, moreover, what kind of rehabilitation they received or what kind of management they received are included.

Sincerely,
